

# Genomic analysis of early transmissibility assessment of the D614G mutant strain of SARS-CoV-2 in travelers returning to Taiwan from the United States of America

Ming-Jr Jian[1], Hsing-Yi Chung[1], Chih-Kai Chang[1], Shan-Shan Hsieh[1], Jung-Chung Lin[2], Kuo-Ming Yeh[2], Chien-Wen Chen[3], Feng-Yee Chang[2], Kuo-Sheng Hung[4], Ming-Tsan Liu[5], Ji-Rong Yang[5], Tein-Yao Chang[6], Sheng-Hui Tang[1], Cherng-Lih Perng[1] and Hung-Sheng Shang[1]

[1] Division of Clinical Pathology, Department of Pathology, Tri-Service General Hospital, National Defense Medical Center, Taipei, Taiwan, Taipei city, Taiwan
[2] Division of Infectious Diseases and Tropical Medicine, Department of Medicine, Tri-Service General Hospital, National Defense Medical Center, Taipei, Taiwan, Taipei city, Taiwan
[3] Division of Pulmonary and Critical Care Medicine, Department of Medicine, Tri-Service General Hospital, National Defense Medical Center, Taipei, Taiwan, Taipei city, Taiwan
[4] Center for Precision Medicine and Genomics, Tri-Service General Hospital, National Defense Medical Center, Taipei, Taiwan, Taipei City, Taiwan
[5] Centers for Disease Control, Taipei, Taiwan, Taipei city, Taiwan
[6] Institute of Preventive Medicine, National Defense Medical Center, Taipei city, Taiwan

Corresponding author
Hung-Sheng Shang,
iamkeith001@gmail.com

## ABSTRACT

**Background**. There is a global pandemic of severe acute respiratory syndrome coronavirus 2 (SARS-CoV-2). Information on viral genomics is crucial for understanding global dispersion and for providing insight into viral pathogenicity and transmission. Here, we characterized the SARS-CoV-2 genomes isolated from five travelers who returned to Taiwan from the United States of America (USA) between March and April 2020.

**Methods**. Haplotype network analysis was performed using genome-wide single-nucleotide variations to trace potential infection routes. To determine the genetic variations and evolutionary trajectory of the isolates, the genomes of isolates were compared to those of global virus strains from GISAID. Pharyngeal specimens were confirmed to be SARS-CoV-2-positive by RT-PCR. Direct whole-genome sequencing was performed, and viral assemblies were subsequently uploaded to GISAID. Comparative genome sequence and single-nucleotide variation analyses were performed.

**Results**. The D614G mutation was identified in imported cases, which separated into two clusters related to viruses originally detected in the USA. Our findings highlight the risk of spreading SARS-CoV-2 variants through air travel and the need for continued genomic tracing for the epidemiological investigation and surveillance of SARS-CoV-2 using viral genomic data.

**Conclusions**. Continuous genomic surveillance is warranted to trace virus circulation and evolution in different global settings during future outbreaks.

# INTRODUCTION

Severe acute respiratory syndrome coronavirus 2 (SARS-CoV-2) is the seventh coronavirus known to infect humans. Among them, SARS-CoV, MERS-CoV, and SARS-CoV-2 can cause severe disease, whereas HKU1, NL63, OC43, and 229E are associated with mild symptoms (*Andersen et al., 2020*; *Gralinski & Menachery, 2020*; *Xie & Chen, 2020*). SARS-CoV-2 is the etiological cause of coronavirus disease 2019 (COVID-19), which was first identified in humans in China in late December 2019 (*Wu & McGoogan, 2020*). The outbreak has since spread further and as of March 2021, there have been 100 million confirmed cases of COVID-19 worldwide with over 2 million deaths (https://www.who.int/publications/m/item/covid-19-weekly-epidemiological-update; access date: 2021/03/08). Fewer than 1000 cases of COVID-19 have been reported on the self-governing island of Taiwan (through March 8, 2021), a number that has remained relatively low due to a series of aggressive containment, quarantine, and monitoring measures that have limited local SARS-CoV-2 transmission (*Chang, 2020*; *Chen et al., 2020*; *Cheng et al., 2020*). The majority of confirmed COVID-19 cases were imported from other countries, including America, European countries, and other Asian countries (*Lin et al., 2020b*).

Mutations arise as a natural occurrence of viral replication. When a newly arising mutation confers a competitive advantage with respect to viral replication, transmission, or escape from immunity, that mutation is maintained in the overall virus population (*Andersen et al., 2020*; *Uddin et al., 2020*). Several SARS-CoV-2 genes have been found to evolve, including those encoding the nucleocapsid (N), viral replicase, and spike proteins (S) (*Dilucca et al., 2020*). The D614G mutation in the spike glycoprotein of SARS-CoV-2 was first detected at a significant level in early March 2020 and spread to global dominance over the following few months (*Al-Zyoud & Haddad, 2021*; *Isabel et al., 2020*; *Islam et al., 2020*; *Korber et al., 2020*; *Zhang et al., 2020*). Up-to-date information on viral genomics is crucial for understanding the global dispersion of SARS-CoV-2 and for providing insight into its pathogenicity (*Furuse, 2021*; *Guan et al., 2020*). Knowing the genome sequence helps us understand how SARS-CoV-2 is mutating into variants and how the virus is passing between people.

Taiwan has experienced two waves of imported COVID-19; the first from China between January to late February 2020, and the second from other countries starting in early March 2020 (*Gong et al., 2020*; *Jian et al., 2021b*). To determine the genetic variations and evolutionary trajectory of the isolates, we compared their genomes to those of global virus strains available through GISAID (*Lo & Jamrozy, 2020*). Here, we characterized the SARS-CoV-2 genome sequences from five travelers who returned to Taiwan from the

United States of America during March and April, 2020. Genome sequences of SARS-CoV-2 were then compared with those deposited in GISAID in order to characterize the genealogical networks and identify possible routes of transmission. Our findings may be used to show how variants of SARS-CoV-2 entered Taiwan, and to determine their association with the implemented quarantine system for entry measures established by the Central Epidemic Command Center (CECC), Taiwan. Our findings have implications for epidemiological investigations and surveillance of viral genomic data.

## MATERIAL AND METHODS

### Collection of clinical specimens

Nasopharyngeal swabs (COPAN's COVID-19 Collection & Transport Kits with Universal Transport Medium or Virus Transport Swabs 147C) were obtained from travelers suspected of having COVID-19. This study was approved by the Institutional Review Board of Tri-Service General Hospital (TSGHIRB No.: C202005041), and was registered on March 20, 2020. Written informed consent was obtained from participants for publication of the case reports. The same IRB NO (TSGHIRB No.: C202005041) used in a previous publication (*Jian et al., 2021b*). We need to clarify the objective is sufficiently different and non-overlapping which the previous publication aimed to investigate the clinical characteristics and differentiation of genetic variation among isolates from a cluster of local familial COVID-19 infection. In this study we addressed different questions with more complex analyses and focus on SARS-CoV-2 genome sequences from travelers who returned to Taiwan from the United States of America. The aim of this study was to trace virus circulation and evolution in Taiwan. Since the D614G mutation in the spike glycoprotein of SARS-CoV-2 surged in early March 2020, variant cases were selected between March to April 2020. Total nucleic acid containing viral RNA was extracted from 0.3 mL of the throat swab supernatant using LabTurbo Viral nucleic acid extraction kits on a LabTurbo 48 AIO automatic extractor (Taigen Bioscience Corp., Taipei, Taiwan). RNA was eluted with 60 $\mu$L of RNase-free water (*Perng et al., 2020*).

### SARS-CoV-2 Real-time reverse transcription-polymerase chain reaction (RT-PCR) testing

Pharyngeal specimens were collected from travelers with symptoms of suspected COVID-19. The presence of SARS-CoV-2 was confirmed by RT-PCR testing according to the guidelines of the Taiwan Center for Disease Control (CDC), as described previously (*Jian et al., 2021a*). Briefly, a SARS-CoV-2 real-time RT-PCR assay using primers, probe, and RT-PCR reagents to detect the *RdRp* and *E* genes of SARS-CoV-2 was performed on an AIO 48 System (LabTurbo, New Taipei City, Taiwan). A diluted viral RNA sample from a COVID-19-positive patient was used as a positive control, which was aliquoted and stored at $-80\,°C$ (a Ct values of $34 \pm 2$ per run was considered acceptable) (*Jian et al., 2021a*). All positive samples were confirmed by the Taiwan CDC Central Laboratory. The five cases (Cases 1–5) were obtained between March and April 2020 and included subjects with a travel history from the USA to Taiwan.

## Whole-genome sequencing of SARS-CoV-2

The Ovation RNA-Seq System V2 (Nugen Technologies, San Carlos, CA, USA) was used to synthesize cDNA, which was then processed into a library described previously (*Gong et al., 2020*). WGS was performed as described previously (*Jian et al., 2021b*). Briefly, whole-genome sequences of the SARS-CoV-2 isolates (TSGH-04, TSGH-08, TSGH-22, TSGH-23, and TSGH-29) were obtained using the Illumina TruSeq Stranded mRNA Library Prep Kit protocol to enrich SARS-CoV-2 cDNA using multiplex RT-PCR amplicons. Next-generation sequencing (NGS) was performed on the NovaSeq 6000 platform (Illumina, San Diego, USA) with paired-end reads.

## Phylogenetic relationship analysis

Phylogenetic analysis was performed as described previously (*Jian et al., 2021b*). Briefly, the SARS-CoV-2 genome sequences from GISAID, and those of the five cases reported herein, were aligned using Clustal Omega (*Sievers et al., 2011*). Aligned nucleic acid sequences were then used to construct phylogenetic trees based on the neighbor-joining tree algorithms provided by PHYLIP (*Felsenstein, 2005*). Using the file created by PHYLIP, the tree was drawn and managed using MEGA X (*Kumar et al., 2018*). Bootstrap analysis was performed with 1,000 replicates and the values were presented adjacent to the branch nodes (*Felsenstein, 1985*).

## Comparative genome sequence analysis and single nucleotide variation (SNV) analysis

The almost full-length genome sequence of SARS-CoV-2 ($\geq$ 29 kb) was retrieved from the GISAID EpiCoV database on February 05, 2021, followed by multiple alignments using MAFFT v7.222. The core regions were compared with those of the Wuhan-Hu-1 genome sequences. The five genome sequences, together with sequences retrieved from GISAID, were aligned using MAFFT. SNV median-joining network analysis was performed using PopART software3 (http://popart.otago.ac.nz.), as previously described (*Sekizuka et al., 2020*). Briefly, early stage (2020/02/18–2020/06/19) SARS-CoV-2 genome sequence from USA were aligned to TSGH-04, TSGH-08, TSGH-22, TSGH-23, and TSGH-29. After further analysis of aligned sequences through CD-HIT, 129 sequences from the USA were selected to BLAST in GISAD, and 3,867 highly similar sequence were applied in a haplotype network analysis combined with COVID-19 sequences from TSGH-04, TSGH-08, TSGH-22, TSGH-23, TSGH-29.

## RESULTS

### Phylogenetic relationship analysis

Data regarding the genome sequences of the five strains obtained in our study (Table 1) were deposited in the GISAID database. To investigate the genomic relationships of these five strains, and to identify the possible sources of infection, we used SARS-CoV-2 sequences from the USA, which spread between 2020/02/18 and 2020/06/19 , deposited in the GISAID database. These sequences were aligned to those of TSGH-04, TSGH-08, TSGH-22, TSGH-23, and TSGH-29. The alignment results were further clustered by

**Table 1  Travel history and basic case information for subjects in this study.**

| Report case | Accession ID | Gender/Age | Travel history | Collection date | SARS-CoV-2 RT-PCR | Ct value |
|---|---|---|---|---|---|---|
| Case 1 TSGH-04 | EPI_ISL_426632 | Male/30 | USA | 03/17/2020 | Positive | 22 |
| Case 2 TSGH-08 | EPI_ISL_427395 | Female/64 | USA | 03/23/2020 | Positive | 14 |
| Case 3 TSGH-22 | EPI_ISL_436107 | Male/25 | USA | 03/31/2020 | Positive | 17 |
| Case4 TSGH-23 | EPI_ISL_436108 | Female/21 | USA | 04/02/2020 | Positive | 23 |
| Case5 TSGH-29 | EPI_ISL_447255 | Female/32 | USA | 03/17/2020 | Positive | 26 |

**Notes.**
CT value, cycle threshold value.

CD-HIT. After cluster analysis, 129 sequences from the USA were selected and aligned. The alignment was used to decipher the time-evolution relationship of the different SARS-CoV-2 genomes. Our data showed that the five strains clustered in different positions of phylogenetic tree (Fig. 1). This indicated that the five imported cases of COVID-19 were likely to have originated from contact with different infection sources in the USA. Further characterization of genetic lineages class shows the five SARS-CoV-2 sequences diversified into lineages B, with multiple sub-lineages B (TSGH-29), B1 (TSGH-04, TSGH-08, and TSGH-23), and B.1.369 (TSGH-22) confirmed by PANGOLIN software (https://pangolin.cog-uk.io/) (Table S1).

## Amino acid variation mapping of SARS-CoV-2 genomes and network analysis

Analysis of the complete genomes of five SARS-CoV-2 strains from patients with a travel history to the USA revealed several missense mutations. The mutations occurred in the S-protein, non-structural protein, and open reading frames (ORFs) (Table 2), including the D614G variant in the S-protein in Cases 1–4; the wild-type form was identified in Case 5. In addition to the S-protein D614G mutation, other missense mutations were identified in NSP12 (RNA dependent RNA polymerase; RdRp) with a P323L mutation, a T85I mutation in NSP2, and a Q57H mutation in ORF3a protein. The genetic changes highlighted above are detailed in Table 2, based on the glue application (http://cov-glue.cvr.gla.ac.uk/) (Table S2). To determine the timing of transmission routes for the five travelers who returned to Taiwan from the USA, whole-genome sequences of the isolates were compared by median-joining SNV network analysis with those of all available SARS-CoV-2 genomes in the GISAID database ($n = 3872$; retrieved Aug 9, 2020). Our findings suggested that these current strains were related to, and clustered chronologically with, other globally identified viruses originally detected in the USA from December 2019 to May 2020 (Fig. 2).

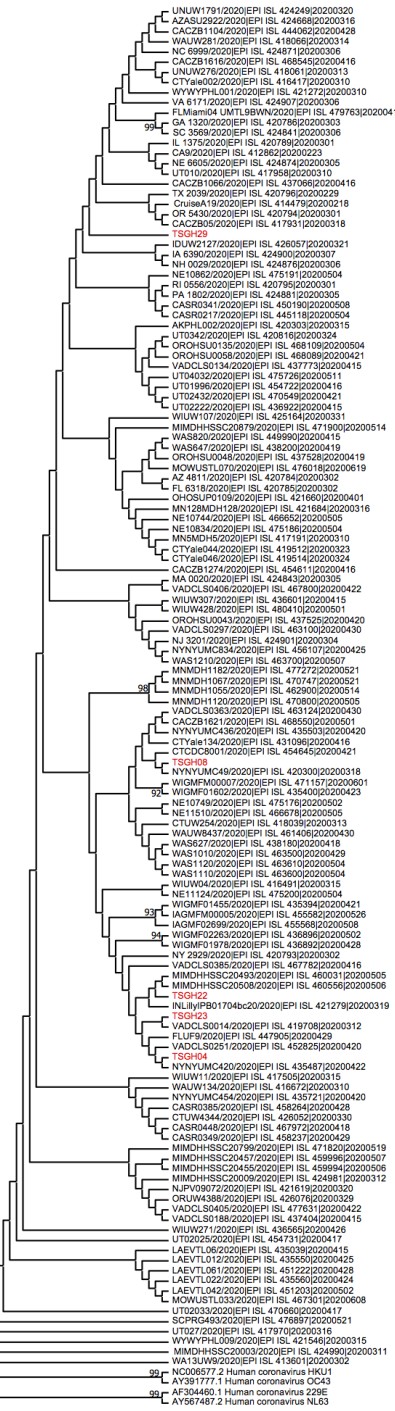

**Figure 1** Phylogenetic analysis of the genomes of 129 strains of SARS-CoV-2 linking the five imported cases of the current study to previously reported genome sequences.

**Table 2  Mutations in the genome sequence of five SARS-CoV-2 strains collected from individuals with a travel history to the United States of America.**

|  | Case 1 | Case 2 | Case 3 | Case 4 | Case 5 |
|---|---|---|---|---|---|
| **Lineage (GISAID clade)**[a] | **GH** | **GH** | **GH** | **GH** | **O** |
| **Best reference hit** |  |  |  |  |  |
| NSP1 hCoV-19/Wuhan/WIV04/2019 | –[b] | – | – | – | – |
| NSP2 hCoV-19/Wuhan/WIV04/2019 | **T85I** | **T85I** | **T85I** | **T85I** | – |
| NSP3 hCoV-19/Wuhan/WIV04/2019 | – | **P1403S** | – | – | **F1141X, R1449G** |
| NSP4 hCoV-19/Wuhan/WIV04/2019 | – | – | – | – | – |
| NSP5 hCoV-19/Wuhan/WIV04/2019 | – | – | – | – | – |
| NSP6 hCoV-19/Wuhan/WIV04/2019 | – | – | – | – | **A161S, I266L** |
| NSP7 hCoV-19/Wuhan/WIV04/2019 | – | **S25L** | **V58F** | – | – |
| NSP8 hCoV-19/Wuhan/WIV04/2019 | – | – | – | – | – |
| NSP9 hCoV-19/Wuhan/WIV04/2019 | – | – | – | – | – |
| NSP10 hCoV-19/Wuhan/WIV04/2019 | – | – | – | – | – |
| NSP11 hCoV-19/Wuhan/WIV04/2019 | – | – | – | – | – |
| NSP12 hCoV-19/Wuhan/WIV04/2019 | **P323L** | **P323L** | **P323L** | **P323L** | – |
| NSP13 hCoV-19/Wuhan/WIV04/2019 | – | – | – | – | **M436L** |
| NSP14 hCoV-19/Wuhan/WIV04/2019 | **L177F** | **A320V** | – | – | – |
| NSP15 hCoV-19/Wuhan/WIV04/2019 | – | – | – | – | – |
| NSP16 hCoV-19/Wuhan/WIV04/2019 | – | – | – | – | – |
| Spike hCoV-19/Wuhan/WIV04/2019 | **D614G** | **D614G** | **D614G** | **D614G** | – |
| NS3 hCoV-19/Wuhan/WIV04/2019 | **Q57H** | **Q57H** | **Q57H** | **Q57H** | – |
| E hCoV-19/Wuhan/WIV04/2019 | – | – | – | – | – |
| M hCoV-19/Wuhan/WIV04/2019 | – | – | – | – | **I8S** |
| NS6 hCoV-19/Wuhan/WIV04/2019 | – | – | – | – | – |
| NS7a hCoV-19/Wuhan/WIV04/2019 | – | – | – | – | – |
| NS7b hCoV-19/Wuhan/WIV04/2019 | – | – | – | – | – |
| NS8 hCoV-19/Wuhan/WIV04/2019 | – | – | – | – | – |
| N hCoV-19/Wuhan/WIV04/2019 | – | – | **S183Y** | – | – |

**Notes.**
[a] Clade and lineage nomenclature was developed by Sebastian Maurer-Stroh et al., based on marker mutations used in GISAID.
[b] no mutations.

# DISCUSSION

Up to March 2021, there had been approximately 1,000 confirmed cases of COVID-19 in Taiwan. Although the number of COVID-19 cases in Taiwan is low compared with other countries, we evaluated five genome sequences of SARS-CoV-2 to understand viral evolution. To gain a better understanding of the genomic epidemiology of the COVID-19 outbreak in Taiwan, we characterized the full genome sequences of five SARS-CoV-2 strains collected from individuals traveling from the USA to Taiwan who were diagnosed with COVID-19 and confirmed to be SARS-CoV-2-positive by RT-PCR. Then, we performed a haplotype network analysis of the SARS-CoV-2 isolates using genome-wide single-nucleotide variations. The D614G SARS-CoV-2 variant appears to have independently arisen and then dispersed throughout multiple geographic regions,

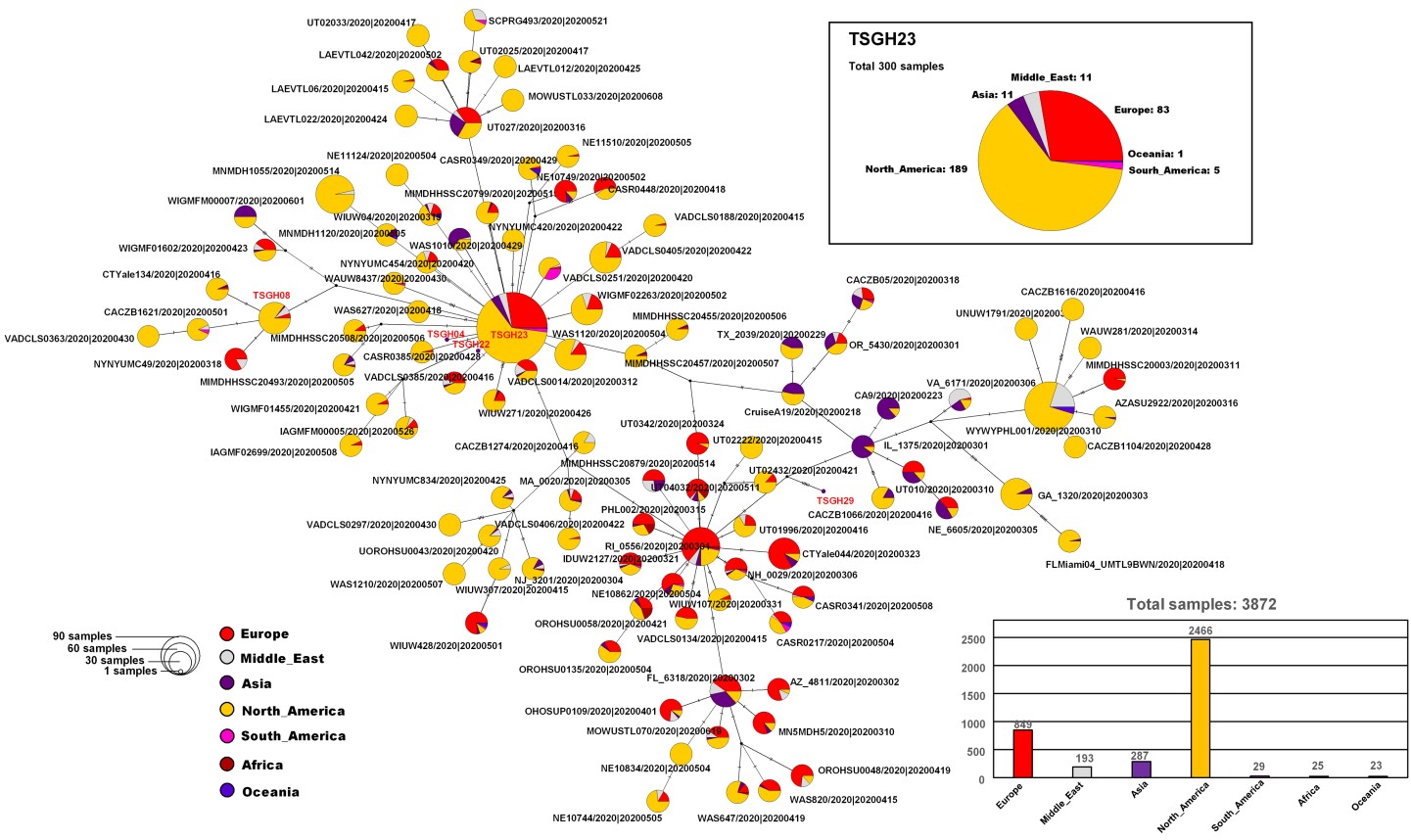

**Figure 2** **Haplotype network established using genome-wide single-nucleotide variations of SARS-CoV-2 isolates.** The haplotype of the SARS-CoV-2 genome sequences for five strains (TSGH-04, TSGH-08, TSGH-22, TSGH-23, and TSGH-29) isolated from five different patients was found to localize in two clusters, which were mainly comprised by North America and European isolates.

including Europe, Latin America, and Asia (*Isabel et al., 2020*; *Islam et al., 2020*; *Kannan et al., 2020*). This is consistent with the timeline of the five COVID-19 cases we evaluated in the current study. Our analysis identified polymorphisms in the S-protein of the current imported cases, which separated into two clusters that were related to globally detected viruses first detected in the USA. Previous studies have shown that the P323L mutation in NSP12 co-evolved with the D614G mutation in S-protein (*Kannan et al., 2020*; *Vilar & Isom, 2021*). Our current findings are consistent with this observation, in that the viral genome sequences of the SARS-CoV-2 strains from Cases 1–4 harbored both the D614G and P323L mutations. Case 5 was found to be wild type for these mutations, and was collected before D614G had become the dominant variant in the world. This demonstrates the need for public health bodies to respond to new viral genetic variants. This is also substantiated by the emergence of the new "UK variant" (lineage B.1.1.7), which began circulating between early September and mid-November, 2020 and has become a variant of widespread concern (*Lauring & Hodcroft, 2021*; *Leung et al., 2021*). More stringent border control and quarantine restrictions have been imposed in response to the emergence of new SARS-CoV-2 variants, such as the UK variant. The spread of this new variant has led to a

surge in COVID-19 cases and deaths (*Davies et al., 2021*). Enhanced genomic surveillance combined with increased compliance with public health mitigation strategies, including vaccination, physical distancing, use of masks, hand hygiene, isolation and quarantine, will be essential to limiting the spread of SARS-CoV-2 and protecting public health (*Lo & Jamrozy, 2020*).

The current study highlights the importance of genomic tracing of the SARS-CoV-2 genome as an epidemiological tool and means of virus surveillance. Travelers have played a significant role in introducing new cases of COVID-19 to countries, perpetuating ongoing human-to-human transmission during the pandemic (*Rodriguez-Morales et al., 2020*). The high transmissibility of SARS-CoV-2, before and immediately after the onset of COVID-19 symptoms, suggests that generalized protective measures may be required, such as social distancing and the wearing of masks (*Cheng et al., 2020*). Taiwan was expected to experience the second highest number of COVID-19 cases due to its proximity to the coast of mainland China. This was avoided due to big data analysis and effective quarantine measures in combination with strong public policies implemented by Taiwan's CECC and CDC. This included the establishment of an entry quarantine system requiring travelers to complete a health declaration form either prior to departure or upon arrival at a Taiwan airport (*Lin et al., 2020a*; *Wang, Ng & Brook, 2020*). Early assessment of airport entry screening has helped Taiwan to detect imported cases of COVID-19, preventing the further spread of the pandemic.

Given the widespread transmission of SARS-CoV-2, additional sequence data regarding new strains are required to fully understand how the virus has spread and evolved. Understanding SARS-CoV-2 variants remains an issue of concern for all governments, and is critical for establishing and implementing effective public health measures. Genomic surveillance of COVID-19 can generate meaningful information for tracking SARS-CoV-2 transmission, and is useful for the real-time response of health departments.

### Funding
This study was supported by the Tri-Service General Hospital, Taipei, Taiwan, ROC (grant number: TSGH-D-110100). The funders had no role in study design, data collection and analysis, decision to publish, or preparation of the manuscript.

### Grant Disclosures
The following grant information was disclosed by the authors:
Tri-Service General Hospital, Taipei, Taiwan, ROC: TSGH-D-110100.

### Competing Interests
The authors declare there are no competing interests.

## Author Contributions

- Ming-Jr Jian conceived and designed the experiments, performed the experiments, analyzed the data, prepared figures and/or tables, authored or reviewed drafts of the paper, and approved the final draft.
- Hsing-Yi Chung, Chih-Kai Chang and Shan-Shan Hsieh conceived and designed the experiments, performed the experiments, authored or reviewed drafts of the paper, and approved the final draft.
- Jung-Chung Lin, Chien-Wen Chen and Feng-Yee Chang conceived and designed the experiments, authored or reviewed drafts of the paper, and approved the final draft.
- Kuo-Ming Yeh and Tein-Yao Chang conceived and designed the experiments, prepared figures and/or tables, authored or reviewed drafts of the paper, and approved the final draft.
- Kuo-Sheng Hung conceived and designed the experiments, prepared figures and/or tables, and approved the final draft.
- Ming-Tsan Liu and Ji-Rong Yang conceived and designed the experiments, analyzed the data, authored or reviewed drafts of the paper, and approved the final draft.
- Sheng-Hui Tang conceived and designed the experiments, analyzed the data, prepared figures and/or tables, and approved the final draft.
- Cherng-Lih Perng conceived and designed the experiments, analyzed the data, authored or reviewed drafts of the paper, and approved the final draft.
- Hung-Sheng Shang conceived and designed the experiments, performed the experiments, prepared figures and/or tables, authored or reviewed drafts of the paper, and approved the final draft.

## Human Ethics

The following information was supplied relating to ethical approvals (i.e., approving body and any reference numbers):

This study was approved by the Institutional Review Board of Tri-Service General Hospital (TSGHIRB No.: C202005041), registered on March 20, 2020. Written informed consent was obtained from the participants for publication of the case report.

## DNA Deposition

The following information was supplied regarding the deposition of DNA sequences:

All viral assemblies are available at GISAID:

TSGH-04 EPI_ISL_426632, TSGH-08 EPI_ISL_427395, TSGH-22 EPI_ISL_436107, TSGH-23 EPI_ISL_436108, TSGH-29 EPI_ISL_447255.

Sequences are also available in the Supplemental Files.

## Data Availability

All viral assemblies are available at GISAID (requires a free account to access): TSGH-04: EPI_ISL_426632; TSGH-08: EPI_ISL_427395; TSGH-22: EPI_ISL_436107; TSGH-23: EPI_ISL_436108; TSGH-29: EPI_ISL_447255

## Supplemental Information

Supplemental information for this article can be found online at http://dx.doi.org/10.7717/peerj.11991#supplemental-information.

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
