# Peer review of "Genomic analysis of early transmissibility assessment of the D614G mutant strain of SARS-CoV-2 in travelers returning to Taiwan from the United States of America"

_PeerJ, doi:10.7717/peerj.11991_

## Round 0.1 · original submission · Major Revisions

Authors should consider all the recommendations of the reviewers to improve the manuscript and data presentation.

Reviewer 1 ·

Basic reporting

In general, the professional English language was used, though some corrections are needed

Experimental design

Would require authors to address some concerns. Please see "General comments for the authors"

Validity of the findings

Would require authors to address some concerns. Please see "General comments for the authors"

Additional comments

In the present study, Jian and colleagues sequenced SARS-CoV-2 genomes from five (5) travelers returning to Taiwan from the United States of America. Although this is an important and interesting study, there are a few areas that I would like the authors to address.

Major comments
1. Line 104-105: The authors should indicate the number of genome sequences retrieved from GISAID and included for the phylogenetic analysis. Based on line 121, it appears that only 37 genome sequences were used for the analysis. Please explain the rationale behind the selection of these 37 sequences. This is crucial because of the 37 sequences, 36 were originated from the USA. Such sequence sampling bias will adversely impact the conclusion. If the total number of sequences included for the analysis was 3,873 (line 139), please include this full phylogenetic tree in Figure 1.
2. Also, in its current form, Figure 1 is not suitable for publication. Please include the accession number for each sequence, the scale, an outgroup, and remove bootstrap values less than 90.
3. Line 123-128: --- “Our data showed that the five strains clustered an extended branch-length distance from the ancestral human SARS-CoV-2 virus, hCoV-19/Wuhan/Hu-1/2019 (denoted as WuhanHu1), but were situated among strains (USA) that are mainly from America (Figure 1). This indicated the five imported COVID-19 cases were likely to have originated from contact with infection sources in the USA, not Taiwan”--- Based on the current Figure 1, such claim is rather contentious because (a) the ancestral Wuhan sequence is not situated at the base of the phylogenetic tree, and (b) the current Figure 1 may also suggest that Wuhan case was originated from the USA. Please address the concern raised in Point #1, and see if the authors' observation remains valid.
4. The authors used genome-wide single-nucleotide variations to establish the haplotype network. This is interesting but would appreciate it if the authors could include a more detailed methodology for this part. The current methodology does not provide sufficient information to enable the readers to reproduce the observation. Also, the sampling bias remains prominent here with 64% of the data originated from the USA.

Other comments
1. Please check the Reference section; The format does not seem to match the requirement.
2. Please include citation(s) or source(s).
(a). Line 55-57: 87% were imported from other countries
(b). Line 70-72
3. Unclear sentence structure
(a). Line 73-76: The SARS-CoV2 genome sequences our five specimens were then compared…”
4. Add “real-time” to Line 82, since the text is more about real-time PCR, not reverse transcription PCR
5. Naming should be “SARS-CoV-2”, not
(a). Line 92: SARS-COV-2
(b). Line 111: SARSCoV-2
6. Table 1: If available, would be nice to include the length of stay of these travelers in the USA.

·

Basic reporting

The article is professional, clear, and informative. I would advise some further English language fluency improvement in order to provide a better reading experience and easier understanding of the content.

Experimental design

1. The method and materials could add more details regarding the method of sample collection and nucleic acid extraction. Even though the article does state the SARS-CoV-2 testing method reference between line 85 to 87, it would help readers have a better understanding of the whole experimental method with some brief description regarding the sample collection and nucleic acid extraction to ensure good sample processing and quality prior to the subsequent qPCR and sequencing.

2. The guideline for variant case selection could be stated here to help the readers understand how the five cases were selected and under what criteria.

Validity of the findings

1. Case 5 (TSGH-29) is significantly closer to Wuhan-Hu-1 strain compared to Case 1, 2, 3, and 4. Please clarify the reasoning of the extended branch-length from the ancestral human SARS-CoV-2 virus.

2. The discussion section mentioned that Case 1 to Case 4 conform with the findings by Kannan et al., 2020; Vilar and Isom, 2021, but did not mention further findings of Case 5, which is quite unique compared to Case 1 to Case 4 providing the mutated genes were completely different. It would be more comprehensive if Case 5 could be further investigated and described.

3. The five cases all had moderate to high viral load. Together with the infection of the mutated SARS-CoV-2, if possible, please describe if the patients have more severe symptoms or special treatments needed compared to the wild type. And if there are any suggestions regarding if the border control should be strengthened or if mandatory variant testing for COVID-19 positive patients should be implemented in order to prevent more deadly variants from spreading to countries based on the clinical significance.

Additional comments

In general, the article provided me extensive information and knowledge regarding the mutated COVID-19 virus. I would appreciate it if the article could also mention deeper insights regarding the importance of preventing the variants from spreading and how the variants could adversely affect public health to provide more supportive facts to the conclusions.

Reviewer 3 ·

Basic reporting

The manuscript by Jian et al. present the analysis of five SARS-CoV-2 samples from patients with a travel history from USA to Taiwan in March and June 2020. The manuscript focuses in the analysis of the D614G change that has been related with a higher transmissibility.

Experimental design

The samples are described as “isolated” but there is no description of culture of the virus. The genomes are obtained using a previous PCR enrichment procedure (ARTIC?) but this is not described in the manuscript. In the current scenario, where there are hundreds of thousand available genome, five genomes seem to be a small sample for analysis. The authors must remark the importance of analyzing this number of samples to understand the evolution of SARS-CoV-2 specific markers.
The analysis of the samples is correct but too basic, a neighbor-joining tree with relatively few numbers of samples, considering the available genomes in the databases, provide scarce results. The D614G change is found in most strains currently circulating around the world and there are several studies that deal with its origin and spreading.

Validity of the findings

no comment

Additional comments

I suggest deepening the classification of the samples using pangolin software (https://pangolin.cog-uk.io/) and to establish the genetic changes using for example the glue application (http://cov-glue.cvr.gla.ac.uk/) that will increase your genome information.

---

## Round 0.2 · Minor Revisions

A few comments from reviewer #1, the authors need to take into account in the work.

Reviewer 1 ·

Basic reporting

I applaud the authors' effort and initiative in addressing all the comments. However, the topology of the phylogenetic tree in Figure 1 does not support the authors' claim. A few matters to consider

1) Phylogenetically speaking, the Wuhan strain does not appear to be an ancestral strain (though a better way to illustrate this may be through genealogical analysis by estimating the tMRCA using BEAST). As of now, it appears that CTYale046/2020 is the ancestral strain, which is contradicting.

2) Lineages clustering in Figure 1: For example, based on Supplementary Table 1, isolate CACZB1066/2020|EPI_ISL_437066|20200416 is categorized as lineage B.46. However, in Fig 1, this isolate was intermingled between TSGH29 and WuhanHu1, both of which were categorized as Lineage B. Please check the clustering and topology of every single isolate in the tree (not just the TSGH isolates), to safeguard the accuracy and reliability of the claim.

3) I am not certain if Hepatitis B virus strain will be a suitable outgroup. I would suggest the authors include the endemic human coronaviruses - OC43, HKU1, NL63, and 229E - as an outgroup.

Experimental design

As mentioned above

Validity of the findings

As mentioned

Additional comments

In general, the work is interesting and important, though some confirmatory analysis is required.

·

Basic reporting

no comment

Experimental design

no comment

Validity of the findings

no comment

Additional comments

The revision of the article addresses all the questions and suggestions with professionalism. No other comments are made.

---

## Round 0.3 · accepted · Accept

The authors have addressed all comments. I support the publication of the manuscript.